# Text-prompt Camouflaged Instance Segmentation with Graduated Camouflage Learning

## ABSTRACT

Camouflaged instance segmentation (CIS) aims to seamlessly detect and segment objects blending with their surroundings. While existing CIS methods rely heavily on fully-supervised training with massive precisely annotated data, consuming considerable annotation efforts yet struggling to segment highly camouflaged objects accurately. Despite their visual similarity to the background, camouflaged objects differ semantically. Since text associated with images offers explicit semantic cues to underscore this difference, in this paper we propose a novel approach: the first **T**ext-**P**rompt based weakly-supervised camouflaged instance segmentation method named TPNet, leveraging semantic distinctions for effective segmentation. Specifically, TPNet operates in two stages: initiating with the generation of pseudo masks followed by a self-training process. In the pseudo mask generation stage, we innovatively align text prompts with images using a pre-training language-image model to obtain region proposals containing camouflaged instances and specific text prompt. Additionally, a Semantic-Spatial Iterative Fusion module is ingeniously designed to assimilate spatial information with semantic insights, iteratively refining pseudo mask. In the following stage, we employ Graduated Camouflage Learning, a straightforward self-training optimization strategy that evaluates camouflage levels to sequence training from simple to complex images, facilitating for an effective learning gradient. Through the collaboration of the dual phases, our method offers a comprehensive experiment on two common benchmark and demonstrates a significant advancement, delivering a novel solution that bridges the gap between weak-supervised and high camouflaged instance segmentation.

## CCS CONCEPTS

• **Computing methodologies → Interest point and salient region detections**.

## KEYWORDS

camouflaged instance segmentation, weakly-supervised, text-prompt

## 1 INTRODUCTION

Camouflaged instance segmentation (CIS) is a task focused on seamlessly detecting and segmenting objects blending with surroundings as instances. The remarkable resemblance of camouflaged objects

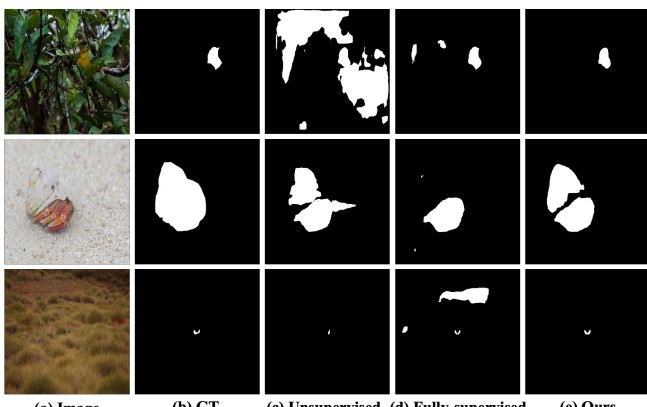

| (a) Image | (b) GT | (c) Unsupervised | (d) Fully-supervised | (e) Ours |

**Figure 1: (a) denotes the original image containing camouflaged objects and (b) signifies the ground truth.Both unsupervised method CutLer [41] (c) and fully-supervised method Mask R-CNN [16] (d) encounter difficulties in distinguishing foreground and background in camouflaged images with similar appearances. To overcome this challenge, we introduce TPNet, a novel approach that leverages text prompts to integrate semantic information and employs graduated camouflage learning for accurate weakly-supervised camouflaged instance segmentation.**

to their surroundings renders the task significantly more difficult compared to generic instance segmentation tasks. Stimulating significant interest within the computer vision community, CIS proves valuable across numerous domains, such as wildlife protection [29, 45], medical image segmentation [14, 15, 22] and industrial defect detection [27].

Recently, benefiting from the strong perceptual capability of deep neural networks, especially the Transformer architecture [39], CIS [11, 26, 31] has made significant progress. However, these methods are all based on fully-supervised learning, which poses challenges due to the difficulty and labor-intensive nature of annotating camouflage images with high precision. In fact, the adoption of weakly-supervised methods to avoid the high costs associated with massive precise annotation required by full supervision has become a trend. This way has been applied in various tasks, such as object detection [43, 48], image segmentation [33], video object segmentation [47], achieving significant progress.

This leads us to consider: Can we explore CIS tasks under weak supervision? Such an approach could significantly alleviate the challenge of accurately labeling camouflage data, which would be of profound significance. However, even fully-supervised trained with precisely labeled data still cannot accurately segment camouflaged objects in CIS. For example, the bird concealed within the

forest in the first row of Figure. 1 is not effectively segmented by Mask R-CNN [16], as its color closely resembles that of the surrounding leaves. Not to mention the disappointing performance of unsupervised methods like Cutler [41]. Due to the lack of prior works conducted under weakly-supervised setting in CIS, therefore it is a significant challenge to introduce appropriate weak supervision to accurately differentiate foreground and background and thus achieve accurate camouflaged instance segmentation.

To this end, we carefully observe the characteristics of the camouflage image itself and then realize that despite the strong resemblance between foreground and background, they differ semantically. As shown in the second row of Figure. 1, although the tail of the snail bears a remarkable resemblance in color and texture to the background beach, it remains clearly distinguishable as a separate semantic instance. Notably, text-prompt have been effectively employed as semantic weak supervision in tasks like camouflaged object detection [18] and open-vocabulary object detection [25], demonstrating remarkable capabilities. These tasks further confirm the feasibility and effectiveness of incorporating semantic weak supervision. Hence, in this paper text-prompt is utilized as weak supervison for the first time to explore CIS.

Besides, some weakly-supervised works in other tasks often utilize self-training methods to refine initial pseudo-labels and obtain final results. These self-training approaches often treat the training data as if it is unordered, implying that samples are introduced into the training process without regard to their difficulty or complexity. However, related researchs [3, 40] on curriculum learning suggest that training with unordered samples does not fully exploit the training potential, while ordered training dataset may enhance model capacity and endow it with stronger generalization ability. Due to the unique characteristics of camouflage images compared to other conventional images, it is feasible to evaluate the camouflage level for each image. Inspired by this, optimizing self-training methods based on graduating the camouflage levels is another key exploration in this paper.

In response to the aforementioned explorations, we propose the first text-prompt based weakly-supervised camouflage instance segmentation framework named TPNet, aimed at alleviating the demand for instance-level pixel annotation. TPNet employs camouflaged text prompts together with images to simultaneously utilize semantic and spatial information. This framework adopts a two-stage approach, initially generating pseudo mask, followed by self-training.

Firstly, during the pseudo mask generation stage, we aim to generate pseudo masks for camouflaged images based on a given text-prompt. We identify camouflaged regions in the image using a existing object detection model and create a series of prompts covering camouflaged categories. Additionally, we align the text-prompt with these instance regions, filtering and pairing them based on cosine similarity. Following this, a Semantic-Spatial Iterative Fusion (SSIF) module is innovatively employed to iteratively refine the pseudo mask. This module integrates semantic and spatial information guided by a carefully designed mask evaluator. In the second stage, we introduce a novel self-training strategy named Graduated Camouflage Learning (GCL). Comprising Camouflage Measurer and Camouflage Scheduler, GCL is devised to utilize the distinctive features of camouflage images to enhance the model's

capabilities. Initially, the Camouflage Measurer evaluates the level of camouflage in images and utilizes this evaluation as the criterion for sorting training samples. Subsequently, the Camouflage Scheduler prioritizes images with lower levels of camouflage during training. As training progresses, more complex images with higher levels of camouflage are gradually incorporated into the training. Through this optimized training strategy, the model can acquire the ability to segment and learn from highly camouflaged samples with low gradient after simple samples with high gradient.

Our major contributions can be summarized as follows:

- We introduce TPNet, the first text-prompt based weakly-supervised framework for camouflaged instance segmentation, which significantly reduces the annotation burden of image data by leveraging semantic understanding for effective segmentation.
- We propose a self-training approach, named Graduated Camouflage Learning (GCL). GCL prioritizes the mastery of simple image features before addressing the challenge of complex camouflage, thereby significantly enhancing the model's accuracy and robustness in handling camouflaged scenes.
- We present Semantic-Spatial Iterative Fusion, a groundbreaking innovation that pioneers the seamless integration of semantic understanding with spatial context. SSIF fuses and refines the pseudo mask through a effective iterative process, yielding refined segmentation results.
- Experimental results indicate that our TPNet outperforms all existing unsupervised and point-supervised instance segmentation models on the CIS dataset, achieving performance comparable to some fully-supervised instance segmentation approaches.

## 2 RELATE WORK

### 2.1 Camouflaged Instance Segmentation

Although Camouflaged Instance Segmentation (CIS) has garnered increasing attention in recent years, it remains a challenging task due to the intricate nature of identifying camouflaged objects in complex backgrounds. Le et al. [20] first introduced the concept of the CIS task and conducted instance-level annotations on the available CAMO dataset, laying the groundwork for CIS research. Pei et al. [31] utilized transformer to address the CIS problem, leveraging its powerful global information processing capabilities to segment camouflaged objects. Building on the work of predecessors, Luo et al. [26] introduced the concept of frequency analysis from the task of camouflaged object detection [9, 46] into CIS, utilizing fourier transform to de-camouflaging. Inspired by query-based transformers [5, 51], Dong et al. [11] propose a unified query-based multi-task learning framework for camouflaged instance segmentation. This framework, by integrating learning objectives from different tasks, further enhance the model's segmentation performance.

Despite the significant achievements of fully-supervised methods in camouflaged instance segmentation (CIS), these methods rely on time-consuming fully-supervised annotations. In particular, annotating camouflaged images becomes notably harder when the foreground and background share striking similarities. However, there has been no exploration of CIS beyond fully-supervised methods. To

tackle this challenge, our work is the first to explore accurately segmenting camouflaged images under a non-fully-supervised framework, effectively reducing the exorbitant costs associated with annotating camouflaged images.

## 2.2 Unsupervised and Weakly-supervised Instance Segmentation

Although there hasn't been any non-fully supervised work on CIS tasks yet, the field of general instance segmentation has seen considerable research. Especially since the advent of deep learning, significant progress has been made. In the unsupervised instance segmentation, DINO [6] has demonstrated that salient features can be acquired through self-supervised learning. Building upon this notion, recent methods such as LOST [36] and TokenCut [44] utilize self-supervised ViT [12] features to segment individual salient objects within images, employing graph-based techniques that leverage DINO's patch features. In weakly-supervised instance segmentation, various types of supervision exist. For instance, some methods [8, 37] employ box supervision, where specialized loss functions are designed to achieve end-to-end instance segmentation. Others [7] utilize point supervision, which, in addition to augment bounding box guidance by introducing a selection of random points, for instance, 3, 5, or 10 points, to refine the prediction of the final mask. Although these methods [38, 41, 42] have been successful in general instance segmentation tasks, their performance in camouflaged instance segmentation tasks is disappointing because of the high similarity between camouflaged objects and their backgrounds. Recently, some research [18, 49] has delved into unsupervised camouflaged object detection, making significant achievements within the field of camouflaged object detection . However, there has been a lack of research focusing on weakly-supervised tasks in CIS. Due to the fundamental differences and distinct requirements between CIS and COD tasks, the methods developed for COD cannot be directly applied to CIS tasks.

In response to these challenges, we introduce the first text-prompt based camouflaged instance segmentation model. Our model does not depend on time-consuming, pixel-level annotations; Instead, it utilizes text prompts from language model as weak supervision. This unique approach leverages rich semantic information to guide camouflaged instance segmentation, overcoming limitations often encountered by general unsupervised methods when dealing with complex scenes.

## 3 METHOD

We introduce TPNet, a novel text-prompt based weakly-supervised framework for camouflaged instance segmentation. The framework primarily consists of two stages: pseudo mask generation and graduated camouflage learning. Firstly, we detail the method's overview (Sec 3.1). Furthermore, we specifically introduce the Semantic-Spatial Iterative Fusion module, a novel component of the first stage that effectively integrates semantic and spatial information to generate refined pseudo masks (Sec 3.2). Additionally, we provide an in-depth discussion on the Graduated Camouflage Learning method, which is in the second stage and designed to optimize the self-training phase of our model, enabling it to handle the complexity of camouflaged images (Sec 3.3).

## 3.1 Overview

Our proposed framework is illustrated in Figure. 2. It primarily consists of two stages: pseudo mask generation and graduated camouflage learning.

In the first stage, given a camouflaged image $I \in \mathbb{R}^{H \times W \times 3}$ and a specific text-prompt $P_s$: "a photo of camouflaged objects", our framework aims to generate pseudo mask $M \in \mathbb{R}^{H \times W \times 1}$. In the image branch, we discover camouflaged regions with a object detection model based on DINO [2] to obtain region proposals $\mathcal{R} = \{r_1, r_2, ..., r_n\}$. To capture more camouflaged instances, we also treat the entire image as a region proposal for prediction. In the text branch, we firstly feed a singular, predefined prompt $P_s$ into GPT [1], which produces a spectrum of prompts $P$ covering diverse camouflaged categories. Then we employ CLIP [32] to align text-prompts and image regions by encoding the regions and text-prompts and compute the cosine similarity metric to filter the region-text pairs.

After obtaining region-text pairs, we innovatively design a Semantic-Spatial Iterative Fusion module to generate the final pesudo mask. Specifically, we employ a Semantic Mask Generator and a Spatial Mask Generator to produce masks that correspond to semantic and spatial features, respectively. Then we iteratively integrate these two kinds of masks to adequately fuse semantic and spatial information and progressively improve the quality of the pseudo mask.

In the second stage, our framework propose a Graduated Camouflage Learning mechanism to train an instance segmentation model based on camouflage level. To clarify, we firstly design a Camouflage Measurer to encode camouflaged images and predefined camouflage level text-prompt, and match each image with camouflage level. After that, we construct a Camouflage Scheduler for graduated training, which gradually increases the camouflage level of the training samples and decreases the gradient of the training loss. Under the collaborative effort of these two stages, our framework leads to the generation of refined CIS results.

## 3.2 Semantic-Spatial Iterative Fusion

In previous studies on unsupervised or weakly-supervised learning, researchers [24, 44] often relied on methods such as spectral decomposition [35] or class activation mapping (CAM) [50] for image segmentation. However, these methods have limitations. For example, spectral decomposition primarily uses spatial information and may not fully utilize semantic cues for segmentation. Similarly, while CAM generates heatmap results to identify relevant image regions, it may not capture fine spatial details effectively. Although a vanilla idea is to simply combine the two kinds of masks using a weighted approach, this method may compromise the quality of generated masks. We will compare this straightforward integration approach in the experiment section.

To this end, we propose a Semantic-Spatial Iterative Fusion (SSIF) module to adequately integrate generated semantic and spatial masks wit CAM and spectral decomposition respectively. As shown in the Figure. 2, SSIF consists of three major components: Semantic Mask Generator, Spatial Mask Generator and Iterative Mask Fusion. SSIF begins by using the Semantic Mask Generator and Spatial Mask Generator to create masks from the image. These masks are

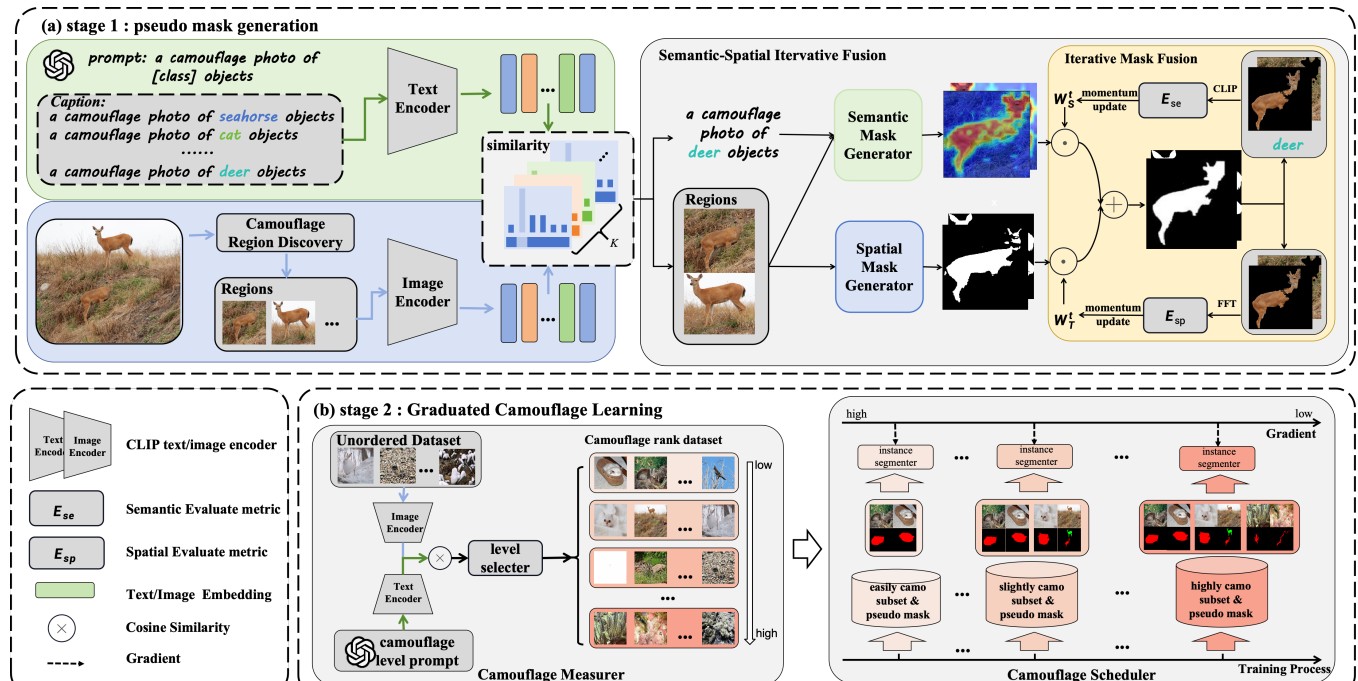

Figure 2: An overview of our proposed TPNet, supervised by camouflage text-prompt. Our framework has two main stages: pseudo mask generation and graduated camouflage learning. In the first stage, using DINO for object detection and GPT for prompt generation, we align text-prompts with image regions to create pseudo masks, refined iteratively by SSIF. In the second stage, we adopt a Graduated Camouflage Learning mechanism to train an instance segmentation model, based on camouflage level.

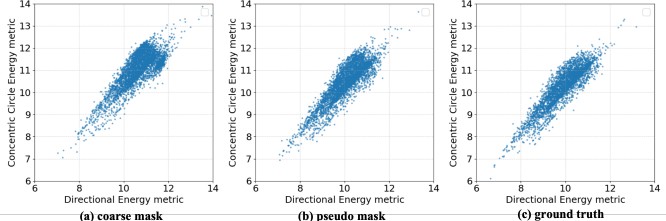

Figure 3: Visualizing the evaluation of different annotations in the training dataset, where each scatter represents an annotation result. It can be observed that the lower the evaluation score, the better the segmentation result matches the instance. (a) shows the evaluation results of the CutLer algorithm ,(b) illustrates our results and (c) presents the evaluation results of ground truth annotations. The x-axis represents energy metric of angular direction on polar coordinates, and the y-axis represents energy metric of radial direction on polar coordinates.

then iteratively refined by the Iterative Mask Fusion component to produce accurate pseudo labels.

**Semantic Mask Generator.** In order to utilize the text prompt, Semantic Mask Generator builds upon the foundation of Class Activation Mapping (CAM) [50], a technique widely used in such tasks to generate heatmaps that highlight regions of interest in an image. It employs feature maps from the final convolutional layer to generate a heatmap. GradCAM [34], on the other hand, improves upon CAM by replacing the weights of the fully connected layer after the Global Average Pooling with gradients to weight the activation map, resulting in a more refined class activation map. Therefore, GradCAM can be effectively employed within the CLIP to generate class activation maps for camouflage image prompts, allowing for the precise localization of object categories within the Vision Transformer model [12].

Given camouflaged region $r \in \mathbb{R}^{h \times w \times 3}$ within single instance and foreground class text prompt $f$, we adopt GradCAM to distinguish foreground class and background class, which can be formulated as:

$$M_{ij}^{se} = \text{RELU}(\sum_k w_k^f A_{ij}^k), \quad (1)$$

where $w_k^f$ denotes the weight of foreground class for k-th feature map, $A_{ij}^k$ denotes the activation value for the k-th feature map and $M^{se}$ denotes the semantic mask and RELU is an activation function that equal to $max(0, x)$.

**Spatial Mask Generator.** We employ DINO and NCut to facilitate the generation of spatial masks. Ncut [35] is a graph-based image segmentation algorithm that utilizes the spectral decomposition of a graph to partition an image into regions of similarity. This is

achieved by solving a generalized eigenvalue system to find the eigenvector $x$ corresponding to the second smallest eigenvalue $\lambda$

$$(D - W)x = \lambda D x, \tag{2}$$

where $W \in N \times N$ is a symmetrical matrix and $D \in N \times N$ is a diagonal matrix derived from $W$. Firstly, we obtain patches yielded by DINO for all regions. Next, we utilize the key feature of each patch to calculate the inter-patch affinity $W$ across all patches within the DINO feature space. This is achieved by solving Eq. 2 to find the second smallest eigenvector $x$. With a threshold value, we obtain the spatial mask $M_{sp}$ using $M^{sp}ij = \max(0, \mathrm{sign}(M^{sp}ij - x))$.

**Iterative Mask Fusion.** After obtaining semantic and spatial masks, we iteratively fuse these two kinds of masks and refine the generated pesudo mask based on the evaluation of fused masks, rather than simply adding them together. Firstly, we evaluate the fused masks on semantic and spatial aspects.

In the semantic aspect, we employ the CLIP model to evaluate the similarity between foreground and background images with the camouflage category text prompt individually. Then, we calculate the ratio between these similarity scores. The entire process can be formulated as:

$$E_{se} = \frac{SIM(x_I^f, x_T)}{SIM(x_I^b, x_T)}, \tag{3}$$

where $x_I^f$ and $x_I^b$ is the feature of the foreground and background mask from image encoder, $x_T$ denotes the feature of camouflaged category text-prompt from text encoder, and $SIM$ denotes the cosine similarity function.

In the spatial aspect, we exploit frequency features to evaluate the spatial quality of fused masks. Despite the strong resemblance between camouflaged objects and the background, disparities in energy distribution exist in the frequency domain.

To visualize the energy distribution across different frequency domains, we conducted an analysis and present the results in Fig. 3. Each point corresponds to a pair of values representing the variance of energy distribution along various frequency axes, with lower values indicating a more balanced distribution. The scatter plot reveals clear distinctions between the ground truth and coarse masks . As shown in Figure. 3 (c), the ground truth, labeled with smaller values on the plot, demonstrate a uniform energy spread, while the coarse masks in Figure. 3 (a) exhibit higher values, pointing to a less balanced distribution. This leads us to conjecture that fully segmented instances present a balanced energy distribution, while incomplete segmentation of camouflaged instances may result in uneven energy distribution, especially where partially disguised regions intersect with the background. Therefore, leveraging frequency features may enable the recognition of subtle differences in spatial quality. In this paper, we introduce Directional Energy Distribution metric $E_{DE} \in [0, 1]$ and Concentric Circle Energy metric $E_{CCE} \in [0, 1]$. These metric respectively calculate the ratio of energy distribution along angular and radial directions on polar coordinates in the frequency domain of the original image $o$ and the foreground $f$ as follows:

$$E_{CCE} = \frac{\ln\left(\sum_{j=1}^M \left(E_j^f - \bar{E^f}\right)^2\right)}{\ln\left(\sum_{j=1}^M \left(E_j^o - \bar{E^o}\right)^2\right)}, E_{DE} = \frac{\ln\left(\sum_{i=1}^N \left(E_i^f - \bar{E^f}\right)^2\right)}{\ln\left(\sum_{i=1}^M \left(E_j^o - \bar{E^o}\right)^2\right)}, \tag{4}$$

where $M$ and $N$ denote the total number of energy values measured along the radial and angle direction respectively, $E_j$ and $E_i$ denote the energy values along angular and radial directions on polar coordinates in the frequency domain, and $\bar{E}$ represents the average energy value. The mask evaluation in spatial aspect is composed of two evaluations as follows:

$$E_{sp} = \beta \cdot E_{CCE} + (1 - \beta) \cdot E_{DE}, \tag{5}$$

where $\beta$ is utilized to balance the contributions of the two evaluation metrics. Guided by mask evaluation results in semantic and spatial aspects, we iteratively fuse two kinds of masks and generate the final pseudo masks. Secondly, the evaluation metrics $E_{se}$ and $E_{sp}$ directly influence the weight of semantic mask and spatial mask during the fusion,. Higher metrics indicate better quality in the corresponding masks, resulting in increased weights. Specifically, we first merge the initial masks generated by spectral decomposition and CAM with fixed weight. In each iteration, we use momentum weighted iterative fusion to update the pseudo mask guided by $E_{se}$ and $E_{sp}$. The whole updating process can be formulated as:

$$W_{se}^{t+1} = (1 - \alpha) \cdot W_{se}^t + \alpha \cdot E_{se}, \tag{6}$$

$$W_{sp}^{t+1} = (1 - \alpha) \cdot W_{sp}^t + \alpha \cdot E_{sp}, \tag{7}$$

$$M^{t+1} = W_{sp}^{t+1} \cdot M_{sp} + W_{se}^{t+1} \cdot M_{se}, \tag{8}$$

where $W_{se}^{t+1}$ and $W_{sp}^{t+1}$ denote weight for the semantic and spatial mask respectively in the next iteration. $\alpha$ denotes momentum parameter controlling the influence of previous iterations' results on the current iteration. $M^{t+1}$ mean the pseudo mask generated in the iteration $t + 1$, and this iterative process is carried out three times to achieve a significantly refined mask.

## 3.3 Graduated Camouflage Learning

It is crucial to note that images with varying level of camouflage within the same dataset exhibit different levels of camouflage. Even for the same semantic instance, the level of camouflage can significantly differ as a function of scene complexity. This variation presents a challenge for the first stage of pseudo mask generation, where the quality of the masks is directly impacted by the degree of camouflage. As illustrated in Fig. 4, image (a) shows a highly camouflaged image where the cat is extremely difficult to discern, resulting in a pseudo mask that poorly captures the object. In contrast, image (f) shows a cat that is not well camouflaged, resulting in a pseudo mask that is comparatively more accurate and easier to generate. In other words, as the camouflage level increases, the difficulty of achieving precise segmentation rises, and the quality of the pseudo masks correspondingly declines.

If these less refined pseudo masks from the first stage were to be directly utilized in the typical self-supervised training setup, they would not fully exploit the potential of the data, potentially leading to suboptimal model performance. This insight has been supported by previous research [3], highlighting the need for a more sophisticated training approach.

Inspired by the human learning process and curriculum learning [3, 40], we propose a graduated camouflage learning mechanism to train an instance segmentation model based on camouflage level, which can acquire the ability to segment and learn from highly camouflaged samples after learning from simple samples. The core

of our proposed mechanism is to gradually introduce more challenging camouflage samples for training once the model is proficient in handling simple samples. This ensures that more difficult samples are introduced to the model only after it has acquired sufficient foundational knowledge. This graduated learning mechanism can help the model better adapt to various degrees of camouflage and improve its performance in different camouflaged situations. To this end, we design two modules: Camouflage Measurer and Camouflage Scheduler for Graduated Camouflage Learning.

**Camouflage Measurer.** While prior works on camouflaged object detection [19, 21] have conducted analyses on the degree of camouflage, they all rely on pixel-level annotations. To quantify the level of image camouflage without mask, we ingeniously propose the Camouflage Measurer, which use CLIP [32] to measure the level of camouflage. We empirically design 6 different formats of CLIP prompts $P_c$ with GPT [1], each representing a different level of quantization. For each image in the camouflaged datasets, we employ CLIP to embed it and compute its similarity scores with each prompt. Subsequently, by identifying the prompt associated with the highest similarity score, we confirm the camouflage level of the image. Formally, this process is represented as:

$$L(I) = \sum_{i=1}^{N} \omega_i \cdot SIM(I, P_c^i), \tag{9}$$

where $\omega_i$ denotes the weight associated with the similarity prompt, and $SIM(I, P_c^i)$ denotes the similarity between the image $I$ and the prompt $P_c^i$.

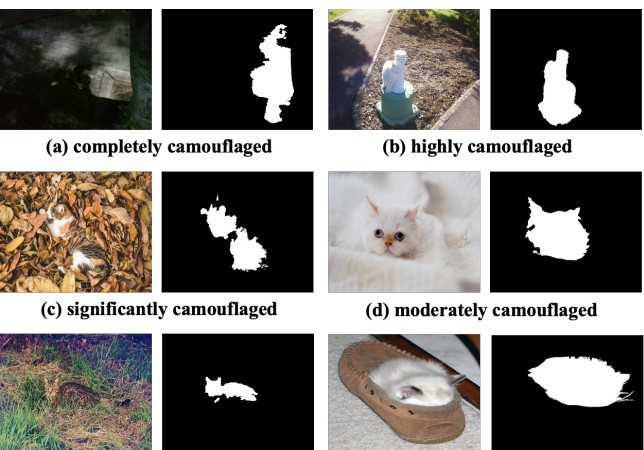

(a) completely camouflaged      (b) highly camouflaged

(c) significantly camouflaged      (d) moderately camouflaged

(e) lowly camouflaged      (f) easily camouflaged

**Figure 4: These six images all depict the same category - a cat, yet with varying level of background camouflage. The images are accompanied by pseudo masks generated in the first stage. From (a) to (e), the cat's level of camouflage progressively intensifies, with (a) being nearly imperceptible. Noting that as the camouflage level increases, the accuracy of the mask decreases.**

**Camouflage Scheduler.** As shown in Figure. 2, the training framework details the workflow of the Camouflage Scheduler. The camouflage levels of all samples are evaluated by the Camouflage Measurer,

and then the samples are sorted in descending order of camouflage level. In the initial stages of training, priority is given to selecting samples with lower camouflage levels, specifically the "easily camo subset and pseudo mask" and "slightly camo subset and pseudo mask", allowing the model to begin learning from easier camouflage instances. For each training batch, the training gradients are adjusted based on the camouflage level of the samples in the batch. Lower camouflage instances are emphasized by receiving higher gradients by increasing learning rate, highlighting the learning of their features. Conversely, higher camouflage instances are allocated lower gradients to mitigate the potential negative impact of inaccuracies in the masks. The loss function can be formulated as:

$$\mathcal{L}_{camo} = L(I) \cdot \mathcal{L}_{ori}, \tag{10}$$

where $I$ denotes the camouflaged image, $L$ denotes the level evaluated by Camouflage Measurer and $\mathcal{L}_{ori}$ denotes original loss result.

As training progresses, the camouflage scheduler gradually adjusts the camouflage levels of samples in the training batches. More complex and highly camouflaged samples, such as those in the "highly camouflaged subset and pseudo mask", are progressively introduced, enabling the model to adapt to increasingly complex camouflage instance. The systematic increase in training with more challenging samples, ensures that the complexity of the learning task gradually escalates, thereby enabling the model to effectively adapt to a broader range of camouflage scenarios.

## 4 EXPERIMENTS

### 4.1 Implementation Details

We utilize four NVIDIA RTX 2080ti GPUs for all experiments and implemented our model using PyTorch[30]. For pseudo mask generation stage, we employ the ViT-B-8 DINO [6] in spatial mask generator and CLIP pre-trained models ViT-B-16 [32] in sematic mask generator. The images are resized to $360 \times 360$ pixels for mask generation. In the self-training process of second stage, we employ Cascade R-CNN [4] for all experiments. To ensure fairness in comparisons, we utilize a ResNet-50 backbone [17] initialized with pre-trained weights from ImageNet [10]. The self-training models are trained for 30,000 iterations with batch size of 8, while the maximum learning rate was set to $2.5 \times 10^{-4}$ and then decays by the step strategy.

### 4.2 Datasets and Evaluation Metrics

**Dataset.** Since CIS is an emerging task and labeling data for it is challenging, datasets for this task are relatively limited. Currently, there are only two publicly available benchmark datasets, COD10K [13] and NC4K [28], which offer instance-level annotations for CIS training. COD10K comprises 5066 images, while NC4K provides instance-level annotations for 4121 images. In this research, we follow the methods and procedures detailed in OSFormer [31] and utilize the COD10K training data as unlabeled images to generate pseudo masks. Additionally, we evaluate our model's performance on both the COD10K and NC4K test sets.

**Evaluation metrics.** We adopt the evaluation metrics of AP50, AP75, and AP scores [23], employing the same settings as used in

**Table 1: Quantitative comparison of our proposed TPNet with state-of-the-art full supervision, point supervision and unsupervision on two benchmark datasets including COD10K and NC4K. The number appended to point-supervised methods typically represents the number of random points utilized.**

| Method | Supervision | Origin | COD10K | | | NC4K | | |
|---|---|---|---|---|---|---|---|---|
| | | | AP | AP50 | AP75 | AP | AP50 | AP75 |
| Mask R-CNN[16] | fully-supervised | CVPR'2017 | 25.0 | 55.5 | 20.4 | 27.7 | 58.6 | 22.7 |
| Cascade R-CNN[4] | fully-supervised | CVPR'2018 | 25.3 | 56.1 | 21.3 | 29.5 | 60.8 | 24.8 |
| OSFormer[31] | fully-supervised | ECCV'2023 | 41.0 | 71.1 | 40.8 | 42.5 | 72.5 | 42.3 |
| UQFormer[11] | fully-supervised | MM'2023 | 45.2 | 71.6 | 46.6 | 47.2 | 74.2 | 49.2 |
| DCNet[26] | fully-supervised | CVPR'2023 | 45.3 | 70.7 | 47.5 | 52.8 | 77.1 | 56.5 |
| PointSup(5)[7] | point-supervised | CVPR'2022 | 17.4 | 43.8 | 11.4 | 17.9 | 45.7 | 11.1 |
| PointSup(10)[7] | point-supervised | CVPR'2022 | 17.9 | 44.1 | 11.9 | 19.1 | 47.6 | 11.6 |
| Tokencut[44] | unsupervised | TPAMI'2023 | 2.6 | 6.5 | 2.0 | 3.5 | 8.3 | 2.5 |
| Freesolo[42] | unsupervised | CVPR'2022 | 12.9 | 37.9 | 6.4 | 16.3 | 46.2 | 7.9 |
| Cutler[41] | unsupervised | CVPR'2023 | 11.7 | 29.1 | 7.3 | 15.5 | 37.9 | 10.5 |
| TPNet | text-prompt | ours | 18.3 | 41.8 | 14.3 | 21.4 | 48.3 | 16.6 |

**Table 2: Ablation study of each component in TPNet on camouflage instance segmentaion.**

| | Method's component | | | | settings on camouflaged instance segmentation | | | | | |
|---|---|---|---|---|---|---|---|---|---|---|
| | | | | | COD10K | | | NC4K | | |
| No. | Baseline | Text prompt | Semantic-Spatial Iterative Fusion | Graduated Camouflage Learning | AP | AP50 | AP75 | AP | AP50 | AP75 |
| ① | ✓ | | | | 11.7 | 29.1 | 7.3 | 15.5 | 37.9 | 10.5 |
| ② | ✓ | ✓ | | | 16.3 | 37.2 | 12.0 | 18.8 | 43.7 | 14.0 |
| ③ | ✓ | ✓ | | ✓ | 16.1 | 37.2 | 11.9 | 19.1 | 44.4 | 14.1 |
| ④ | ✓ | ✓ | ✓ | | 18.2 | 41.7 | 14 | 21.0 | 48.3 | 15.3 |
| ⑤ | ✓ | ✓ | ✓ | ✓ | **18.3** | **41.8** | **14.3** | **21.4** | **48.3** | **16.6** |

OSFormer [31] to evaluate the segmentation results. These metrics is used to evaluate the performance of a model at different Intersection over Union (IoU) thresholds.

## 4.3 Experiment Results and Analysis

To validate the effectiveness of TPNet, we compare various supervision settings for camouflaged instance segmentation, including unsupervised, point supervision, and full supervision, as depicted in Table. 1. The considered methods encompass general unsupervised approaches utilizing Tokencut [44], FreeSolo [42] and Cutler [41]; general point-supervised method employing PointSup [7]; fully-supervised methods such as Mask R-CNN[16], cascade R-CNN [4], OSFormer [31], UQFormer [11] and DCNet [26]; and our proposed weakly-supervised approach TPNet. All methods or results are obtained from official downloads.

Based on the experimental results presented in Table. 1, our method, TPNet, comprehensively outperforms existing unsupervised and point-supervised methods in terms of segmentation performance on both datasets. Specifically, our results demonstrate a significant improvement over the best-performing unsupervised instance segmentation model Freesolo [42], with a 41% increase in AP on the COD10K dataset and a 31.3% increase in AP on the NC4K dataset. Furthermore, when compared to point supervision methods, our approach achieves a 19.6% increase in AP. While there is still a gap when compared to fully-supervised methods,

**Table 3: Quantitative comparison of our proposed SSIF with simple fusion.**

| Method | COD10K | | | NC4K | | |
|---|---|---|---|---|---|---|
| | AP | AP50 | AP75 | AP | AP50 | AP75 |
| w/o SSIF | 16.3 | 37.2 | 12.0 | 18.8 | 43.7 | 14.0 |
| w/ fusion | 15.4 | 35.5 | 11.7 | 18.5 | 42.8 | 13.5 |
| w/ SSIF | 18.3 | 41.8 | 14.3 | 21.4 | 48.3 | 16.6 |

TPNet's performance is notably close to that of Mask R-CNN [16]. This demonstrates that TPNet effectively bridges the gap between weak supervision and the high accuracy required for camouflaged instance segmentation.

## 4.4 Ablation Studies

We conduct ablation studies to validate the respective roles of proposed components in the overall framework.

**Effect of text prompt.** To validate the effectiveness of text prompts, we first establish the DINO encoder and cascade R-CNN in self-training as our baseline, denoted as ①. Building upon this foundation, we incorporate the text branch with text prompt, setting this configuration as ②. Compared to baseline ①, it is evident that the introduction of text prompt in ② leads to a significant performance improvement, achieving a 10% increase in AP. This further demonstrates the effectiveness of semantic information in CIS.

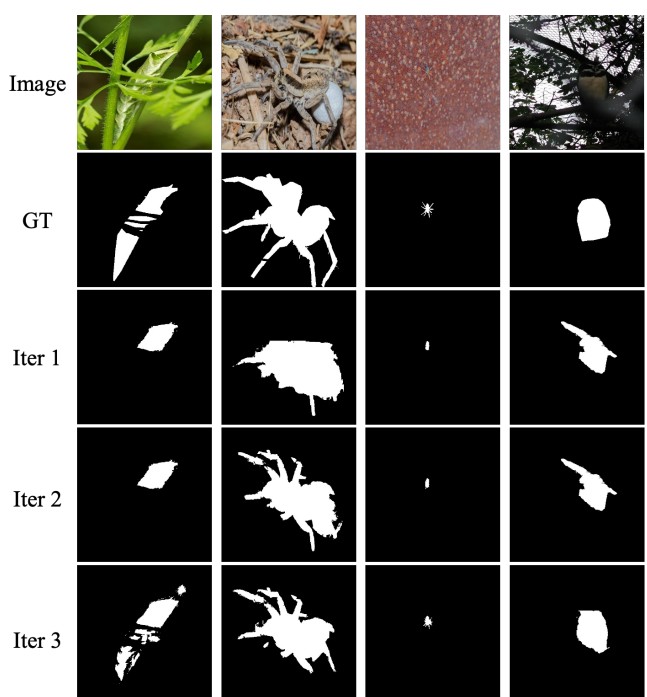

Image

GT

Iter 1

Iter 2

Iter 3

**Figure 5: Visual comparison of different iterative qualitative results. The third to fifth lines demonstrate the results after iterative fusion for 1 to 3 iterations.**

**Effect of Semantic-Spatial Iterative Fusion.** Firstly, we conduct a comparison between a segmentation method that excludes semantic information and a basic vanilla weighted fusion approach, as shown in Table 3. The results indicate that the simple weighted method falls short in accurately segmenting camouflaged instances. This indicates that in camouflaged instance segmentation, direct fusion may not fully leverage semantic information.

To better integrate spatial information with semantic insights, then we employ our proposed method SSIF based on the setting of ②. The results are shown in the rows of ④. Compared results of the setting of ④ with the setting of ②, we can find significant improvements with an increase of 11.16% and 12% on COD10K in terms of AP and AP50, respectively. The significant improvement in quality observed with SSIF demonstrates its effectiveness in identifying and leveraging rich semantic and spatial features to enhance mask results.

We also verify the effectiveness of iterative fusion technique in SSIF. As shown in Fig. 5, we conduct experiments with one, two, and three iterations, and observe a significant improvement in the quality of camouflaged instance segmentation results as the number of iterations increased. Particularly, with three iterations, we find that the details and edges of the images were segmented more clearly, confirming the effectiveness of the iterative fusion method in enhancing image quality. Therefore, we choose three iterations as our final approach.

**Effect of Graduated Camouflage Learning.** To assess the impact of GCL on model performance, we implement GCL as a modification to the original self-training process based on ② and ④. The

**Table 4: Result on varying level of camouflaged dataset sampled from COD10K.**

| camouflage level of test dataset | AP | AP50 | AP75 |
|---|---|---|---|
| highly camouflage | 16.9 | 36.1 | 12.9 |
| moderately camouflage | 17.3 | 39.2 | 12.4 |
| lowly camouflage | 19.5 | 49.4 | 12.4 |

outcomes of this experiment are detailed in ③ and ⑤. ⑤ represents the complete combination of our method. GCL, through graduated learning on training images based camouflage levels, acquires the ability to segment challenging samples. Our ablation experiment results clearly demonstrate the significant improvement in accuracy and robustness achieved by the GCL module.

**Effect of camouflage level.** To assess the influence of varying degrees of camouflage on segmentation accuracy, we conduct a validation experiment by sampling three subsets from the COD10K test dataset based on different levels of camouflage: highly camouflaged dataset, moderately camouflaged dataset, and lowly camouflaged dataset. In Table. 4, as the level of camouflage increases, all metrics decline accordingly. This validation experiment not only validates the effectiveness of our Camouflage Measurer but also emphasizes that images with higher levels of camouflage yield worse segmentation results.

## 4.5 More Consideration

While TPNet shows promising results, challenges remain in the quality of image prompt generated and current unsupervised object region detection. Firstly, the quality of image propmt generated presents a significant challenge, directly impacting the efficacy of text prompt-supervised learning. Improving the generation quality of image descriptions is imperative for advancing the performance of our approach. Secondly, the limitations of current unsupervised object region detection frameworks, particularly the lack of fine-tuning, pose challenges in accurately identifying camouflaged objects. This limitation adversely affects the overall performance of our model. Overcoming these bottlenecks necessitates innovative approaches to enhance image description generation and fine-tuning strategies for unsupervised object region detection frameworks. We will do more in-depth research in the future to meet these challenges.

## 5 CONCLUSION

In this study, we proposed TPNet, the first text prompt based framework for camouflaged instance segmentation, aiming to leverage both visual and semantic information from images and text streams for camouflaged instances mask. In the pseudo mask generation and self-training stages, we introduce the Semantic-Spatial Iterative Fusion (SSIF) and Graduated Camouflage Learning (GCL) modules respectively. SSIF assimilate spatial information with semantic insights, iteratively refining pseudo mask guided by the mask evaluator. Additionally, we introduce GCL, a self-training strategy that uses images of different camouflage levels to establish a gradient affected by the level of camouflage to overcome the accuracy issues caused by camouflage images. Experimental results demonstrate that our proposed network achieves outstanding performance on two common benchmark.

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
