# OpenReview forum: "Text-prompt Camouflaged Instance Segmentation with Graduated Camouflage Learning"
_acmmm.org/ACMMM/2024/Conference — MM2024 Poster_

### Official Review · Reviewer_9FbE · 2024-05-19

**Rating:** 5
**Confidence:** 4

**Summary:**

The authors introduce a framework TPNet for the text-prompted weak-supervised camouflage instance segmentation. Only class labels are utilized during training for supervision. Text prompts together with camouflaged regions detected by DINO are used to generated pseudo masks with CLIP. They also designed a camouflage measurer to predict the camouflage scores and for the Graduated Camouflage Learning process.

**Strengths:**

It is the first Text-Prompt based weakly-supervised camouflaged instance segmentation method. Authors combine the text prompts and camouflage region discovery cleverly for Text-Prompt based weakly-supervised CIS.
The camouflage evaluation method and the Energy metric for evaluation are interesting.

**Limitations:**

1. In Figure 2, both camouflage regions and text prompts are fed into the Semantic Mask Generator for heatmap generation. It is not clearly explained in the 'Semantic Mask Generator' section how they are used. The authors mentioned 'effectively employed within the CLIP'. It's confusing as the process is not well illustrated.
2. What is the author's rationale for designing the different formats of CLIP prompts in 6, and what is the criteria for grading the different levels of quantization? These key questions are not clearly stated in Section 3.3.
3. In table 2, the results with/without Graduated Camouflage Learning are almost the same. I doubt whether this module is useful or not.
4. The authors mention that they employ the ViT-B-8 DINO in the spatial mask generator and CLIP pre-trained models ViT-B-16 in the semantic mask generator. Both are heavy models, which raises the concern about the computational complexity of the method. Could the authors add comparisons of inference time and GFLOPS with previous models?
5. The model performance in Table 1 is not good enough, which raises concerns about model effectiveness. If possible I would suggest that the author could make the code public after the final revision. Also, in Table 1, the AP scores for Tokencut, a method accepted by TPAMI 23', are below than 5. I doubt whether the authors have implemented this method correctly or not. Could the authors provide possible reasons for this?
6. A lot of writing errors in the manuscript, for example:
Implementation Details: ..., while the maximum learning rate was set to 2.5 × 10−4 and then decays by the step strategy. --> decay?
Evaluation metrics: These metrics is used to ... --> are used to?
Experiment Results and Analysis: Table.1 or Table 1? Fig. X or Figure. X?
Table 1: ECCV’2023-->ECCV’2022.

**Suitability:**

3

---

### Official Review · Reviewer_U7oG · 2024-05-27

**Rating:** 5
**Confidence:** 3

**Summary:**

The paper "Text-prompt Camouflaged Instance Segmentation with Graduated Camouflage Learning" introduces TPNet, a novel text-prompt based weakly-supervised framework for camouflaged instance segmentation (CIS). The primary aim is to detect and segment objects that blend into their surroundings using minimal annotations. TPNet operates in two stages: pseudo mask generation and graduated camouflage learning. The pseudo mask generation stage aligns text prompts with images using a pre-trained language-image model, creating pseudo masks iteratively refined through a Semantic-Spatial Iterative Fusion (SSIF) module. In the self-training stage, Graduated Camouflage Learning (GCL) sorts training images by their camouflage level, facilitating an effective learning gradient from simpler to more complex images. The method significantly reduces the annotation burden while achieving superior segmentation performance on benchmark datasets compared to existing unsupervised and point-supervised methods.

**Strengths:**

•	Novelty: The paper presents the first text-prompt based weakly-supervised framework for CIS, addressing the significant challenge of annotating camouflaged images.

•	Methodological Innovation: TPNet combines semantic and spatial information through a Semantic-Spatial Iterative Fusion (SSIF) module, enhancing the refinement of pseudo masks.

•	Effective Self-Training: The Graduated Camouflage Learning (GCL) strategy optimizes self-training by progressively introducing images with higher camouflage levels, improving the model's robustness and accuracy.

•	Practical Applications: The approach is applicable across various domains, including wildlife protection, medical image segmentation, and industrial defect detection.

•	Evaluation: The experimental results demonstrate TPNet's superior performance over existing unsupervised and point-supervised methods, achieving results comparable to fully-supervised approaches.

**Limitations:**

•	Dependency on Text Prompts: The method's reliance on text prompts requires the creation of diverse and accurate prompts, which may be challenging and could impact the model's performance if not done correctly.

•	Complexity of Implementation: Integrating SSIF and GCL adds complexity to the implementation, potentially making it difficult for practitioners to adopt without extensive understanding.

•	Generalization to Diverse Contexts: While the method shows promising results on benchmark datasets, its performance in more diverse and unstructured real-world scenarios remains to be extensively validated.

•	Evaluation Metrics: The paper primarily compares TPNet with unsupervised and point-supervised methods. Additional comparisons with other weakly-supervised approaches could provide a more comprehensive evaluation of its effectiveness.

**Suitability:**

3

---

### Official Review · Reviewer_shNc · 2024-05-30

**Rating:** 2
**Confidence:** 3

**Summary:**

-The paper addresses camouflaged instance segmentation by proposing TPNet, a weakly-supervised framework utilizing text prompts and self-learning method. The text-prompt information is claimed to enhance the camouflaged features. The graduated camouflage learning method benefits the learning process of the model by gradually increasing the level of camouflaged instances.
- Despite the contributions, the performance of the proposed method is limited.

**Strengths:**

- The paper explores the text prompt to enhance camouflaged features, which provide more information to distinguish camouflaged instances and background.
- The paper is among the first works addressing CIS in a non-fully-supervised manner.
- The paper is well-organized and easy to follow.

**Limitations:**

- The storytelling in the introduction lacks comprehension, it is recommended that the authors provide further detailed explanation behind the utilization of text-prompt and self-learning in camouflage research.

- It is recommended that the authors provide further discussion on the general text-prompt-based and self-training approach in the Related Work.

- The proposed methods adopt a combination of existing models to address the task, thus the contributions appear marginal. The limitation is that the authors did not provide the analyzed reasons behind each of their proposed adoptions, e.g. CAM/GradCAM in Semantic Mask Generator or DINO/NCut in Spatial Mask Generator.
- The authors decide to iteratively fuse the two semantic and spatial information to form the pseudo mask due to the failure of weighted fusion. Please provide further explanation on how they are weighted fused.


- As indicated in Table 2, the proposed GCL seems to contribute minor improvement in comparison with the other components, i.e. (4) vs (5). Please clarify the contribution of the GCL in the framework.
- In GCL - Camouflage Measurer, the authors select CLIP [32] to measure the level of camouflage. Why do the authors believe that the CLIP model can well extract the camouflage features?
- How many levels of camouflage are effective in the proposed method?

Misc.:
- Please proofread the manuscript to correct existing typos or paragraph re-arrangements (e.g. paragraph in L493, L569).

**Suitability:**

2

---

### Official Review · Reviewer_xhGN · 2024-06-02

**Rating:** 4
**Confidence:** 3

**Summary:**

The authors propose a new method called TPNet for camouflaged instance segmentation (CIS). TPNet uses text associated with images to help find hidden objects more effectively.

TPNet works in two stages: first, it generates rough outlines of objects based on text cues and images, and then it refines these outlines using a clever method that combines spatial and semantic information. Additionally, TPNet employs a self-training process that gradually learns to recognize more difficult camouflage patterns, improving its performance over time.

Experiments show that TPNet performs better than existing point-supervised or unsupervised methods, making it a promising approach for weakly-supervised camouflaged instance segmentation.

**Strengths:**

The idea of using text to improve the segmentation results of camouflaged objects is interesting and promising.
TPNet innovatively integrates text cues with images, improving segmentation accuracy. The Semantic-Spatial Fusion refines outlines iteratively, enhancing segmentation quality. Graduated Camouflage Learning optimizes self-training, adapting to varying camouflage complexities. Extensive benchmarking shows TPNet outperforms existing methods reliably.

**Limitations:**

As the authors already pointed out, TPNet's performance may be limited without high-quality prompts, highlighting the need for improvements in image description generation.
The manuscript should be revised to correct spelling or grammatical errors.

**Suitability:**

2

---

### Meta-Review · Area_Chair_vKk8 · 2024-07-01

**Recommendation:** Accept (Poster)
**Confidence:** 5

**Metareview:**

This paper has received two WAs, one BA, and one WR as initial scores.

Pros:
All reviewers recognize the novelty about using the text prompt to improve the segmentation results of camouflaged objects. The experimental results demonstrate TPNet's superior performance over existing unsupervised and point-supervised methods.

Cons:
Reviewers xhGN and U7oG have concerns about the quality of the prompts that may affect the performance of the proposed TPNet.
Reviewers shNc and 9FbE raise questions about the model performance as shown in Table 1 and Table 2.

The authors provide a rebuttal which addresses the reviewers' concerns. The AC agrees with the majority of reviewers that the paper should be accepted.